

# Immune responses in carp strains with different susceptibility to carp edema virus disease

Ali Asghar Baloch[1], Dieter Steinhagen[2], David Gela[1], Martin Kocour[1], Veronika Piačková[1] and Mikolaj Adamek[2]

[1] Faculty of Fisheries and Protection of Waters, South Bohemian Research Center of Aquaculture and Biodiversity of Hydrocenoses, University of South Bohemia in České Budějovice, Vodňany, Czech Republic

[2] Fish Disease Research Unit, Institute for Parasitology, University of Veterinary Medicine Hannover, Hannover, Germany

## ABSTRACT

Carp edema virus disease (CEVD), also known as koi sleepy disease (KSD), represents a serious threat to the carp industry. The expression of immune-related genes to CEV infections could lead to the selection of crucial biomarkers of the development of the disease. The expression of a total of eleven immune-related genes encoding cytokines (IL-1$\beta$, IL-10, IL-6a, and TNF-$\alpha$2), antiviral response (Mx2), cellular receptors (CD4, CD8b1, and *GzmA*), immunoglobulin (IgM), and genes encoding-mucins was monitored in gills of four differently KSD-susceptible strains of carp (Amur wild carp, Amur Sasan, AS; Ropsha scaly carp, Rop; Prerov scaly carp, PS; and koi) on days 6 and 11 post-infection. Carp strains were infected through two cohabitation infection trials with CEV genogroups I or IIa. The results showed that during the infection with both CEV genogroups, KSD-susceptible koi induced an innate immune response with significant up-regulation ($p < 0.05$) of IL-1$\beta$, IL-10, IL-6a, and TNF-$\alpha$2 genes on both 6 and 11 days post-infection (dpi) compared to the fish sampled on day 0. Compared to koi, AS and Rop strains showed up-regulation of IL-6a and TNF-$\alpha$2 but no other cytokine genes. During the infection with CEV genogroup IIa, Mx2 was significantly up-regulated in all strains and peaked on 6 dpi in AS, PS, and Rop. In koi, it remained high until 11 dpi. With genogroup I infection, Mx2 was up-expressed in koi on 6 dpi and in PS on both 6 and 11 dpi. No significant differences were noticed in selected mucin genes expression measured in gills of any carp strains exposed to both CEV genogroups. During both CEV genogroups infections, the expression levels of most of the genes for T cell response, including CD4, CD8b1, and *GzmA* were down-regulated in AS and koi at all time points compared to day 0 control. The expression data for the above experimental trials suggest that both CEV genogroups infections in common carp strains lead to activation of the same expression pattern regardless of the fish's susceptibility towards the virus. The expression of the same genes in AS and koi responding to CEV genogroup IIa infection in mucosal tissues such as gill, gut, and skin showed the significant up-regulation of all the cytokine genes in gill and gut tissues from koi carp at 5 dpi. Significant down-regulation of CD4 and *GzmA* levels were only detected in koi gill on 5 dpi but not in other tissues. AS carp displayed significant up-expression of Mx2 gene in all mucosal tissues on 5 dpi, whereas in koi, it was up-regulated in gill and gut only. In both carp strains, gill harbored a higher virus load on 5 dpi compared to the other tissues. The results showed that resistance to CEV could not

Corresponding author
Veronika Piačková,
piackova@frov.jcu.cz

be linked with the selected immune responses measured. The up-regulation of mRNA expression of most of the selected immune-related genes in koi gill and gut suggests that CEV induces a more systemic mucosal immune response not restricted to the target tissue of gills.

# INTRODUCTION

Common carp rearing condition has intensified in recent decades, which could have resulted in increased chronic stress and a weakened immune system, increasing their susceptibility to various pathogens. Among them, infection caused by carp edema virus (CEV) is gaining attention, as it is characterized by high mortality rates but can be also manifested as subclinical infection (*Amita et al., 2002*; *Ono, Nagai & Sugai, 1986*). Therefore the disease caused by CEV and known as carp edema virus disease (CEVD), or koi sleepy disease (KSD) could represent a significant threat to the carp industry. The KSD name has been derived from the main clinical signs manifested by affected fish which usually show lethargy and increasing unresponsiveness, as they can be seen lying in the bottom of tanks for extended periods (*Pretto et al., 2013*). Gross lesions incorporating spreading hemorrhagic skin lesions with edema, particularly in the abdomen, pale gills, sunken eyes, and ulcerative inflammation on the anus may also be seen (*Oyamatsu et al., 1997*). A large amount of mucus is produced on the skin and gills as well (*Zhang et al., 2017*). The CEVD/KSD was first reported in koi farms in Japan in 1974, where it caused substantial mortalities (*Amita et al., 2002*; *Ono, Nagai & Sugai, 1986*) and economic losses. Since then, the disease has spread almost worldwide (*Machat et al., 2021*).

Carp edema virus (CEV), which belongs to the Poxviridae family, has a mulberry-like structure made up of double-stranded DNA about 250–280 nm in diameter (*Oyamatsu et al., 1997*). The infection seems to be most prevalent in the gills. An electron microscope revealed that diseased fish exhibit morphologically altered gill epithelium that displays poxvirus-like structures (*Jung-Schroers et al., 2015*; *Miyazaki, Isshiki & Katsuyuki, 2005*; *Haenen et al., 2014*; *Pretto et al., 2013*). Two to three different genogroups (genetic clades) of CEV have been so far characterized: I, IIa, and IIb (*Way & Stone, 2013*; *Matras et al., 2017*; *Adamek et al., 2017b*). Common carp are the primary hosts of genogroup I, which has been detected in most European waters. A majority of genogroup IIa reports have been in koi, but not exclusively, while genogroup IIb has been detected in both carp and koi samples (*Matras et al., 2017*; *Adamek et al., 2018*; *Matějíčková et al., 2020*; *Ouyang et al., 2020*).

Like higher vertebrates, teleosts have an immune system that employs both specific (adaptive) and non-specific (innate) responses against pathogens such as viruses, bacteria, and parasites (*Whyte, 2007*). The non-specific immune response is considered the very first

defence against pathogens. Unlike other vertebrates, fish primarily rely on the non-specific immune system for survival during early embryonic development (*Rombout et al., 2005*). The fact that fish are poikilothermic also means they rely less on some of the conventional characteristics of adaptive immunity, due to slow and relatively low antibody defences and slow proliferation, maturation, and memory of lymphocytes. Thus, it is considered that innate immune responses play a fundamental role in fish immunity.

In higher animals infections with poxviruses are contained through combination of innate and adaptive immunity. The response to infection is primary undertaken by the inflammatory and natural killer (NK) cells (*Smith & Kotwal, 2002*). These non-specific responses control viral replication and allow the time for mounting a specific antigenic adaptive response (*Magnadottir, 2010*). For instance, T lymphocytes, also referred to as T cells, are a crucial component of the adaptive immune response to infections. They are a type of white blood cell that have the ability to recognize and respond to viral antigens and regulate the immune response through cytokine production (*Andersen et al., 2006*; *Tortorella et al., 2000*). Research has indicated that T cell responses play a vital role in controlling pox virus infections, and the activation and expansion of T cells is a significant factor in resolving the infection (*Yamaguchi et al., 2019*). Furthermore, genetic variations in fish have been shown to affect the T cell response magnitude and specificity to pox viruses, which may impact the outcome of the infection (*Adamek et al., 2021*). During CEV infection in carp the immune response is somehow complicated by onset of environmental immunosuppression caused by intoxication with ammonia (*Adamek et al., 2021*), however the host–virus interaction is reciprocally impacted by host genetic competitiveness and virus genomic characteristics. Several emerging genogroups of CEV have so far been identified among carp populations worldwide. These genogroups are seemingly selective mutations targeted toward the numerous variants within the common carp strains/species (*Adamek et al., 2017c*). It has been also reported, that there are differences between various strains of carp in the expression of cytokines during viral infections, specifically the interferon (IFN) and interferon-stimulated genes (ISGs) (*Tadmor-Levi et al., 2019*). Several varieties of common carp have emerged as an outcome of natural geographic separation of common carp groups and domestication, providing a wide range of genetic resources. As already has been published, there may be a large variations in susceptibility among strains that have a different genetic background, as was evident in studies focused on evaluation of susceptibility of different common carp strains to experimental infection of cyprinid herpesvirus 3 (CyHV-3) (*Shapira et al., 2005*; *Rakus et al., 2009*; *Piačková et al., 2013*; *Adamek et al., 2019*). Comparatively to other viral diseases in fish, scant studies have been conducted on the gene expression patterns of immune-related genes in CEV-infected carp strains. In case of CEV, researchers have exposed different strains of carp (Amur wild carp, Amur sasan, AS; Ropsha scaly carp, Rop; Prerov scaly carp, PS; and koi), having different susceptibility to CEV, genogroups I and IIa (*Adamek et al., 2017a*). However, the study focused primarily on determining whether carp strains were susceptible to the disease, and only type I interferon responses as the parameter of non-specific immunity were assessed for CEV affected carp strains. The mucosal-epithelial barrier is a vital component of the innate immune response of fish against viruses (*Langevin et al., 2013*; *Secombes & Zou, 2017*). It
includes the gills, skin, and gastrointestinal tract as its main components (*Huttenhuis et al., 2006*; *Magnadottir, 2010*), all of which encounter pathogenic agents present in the aquatic environment. This barrier employs various mechanisms to hinder pathogen invasion, including the release of antimicrobial factors by immune cells or tissues. The gut mucosal layer serves as the first barrier against pathogen invasion, but it also provides nutrients for bacterial pathogens (*Garcia et al., 1997*), resulting in a delicate balance between gut mucosal secretions and microorganisms. Studies on innate immune responses of fish gut epithelial cells against viral infections are limited. Carp gut epithelial cells exhibit increased expression of cytokines such as IFN-$\alpha$2, IL-1 $\beta$, and iNOS upon cyprinid herpes virus 3 infection, highlighting the significance of gut epithelial cytokine signalling in maintaining mucosal immunity (*Syakuri et al., 2013*). The skin serves as the primary line of defence against invading pathogens and plays a critical role in immune responses. However, the molecular mechanisms underlying the fish skin's immune response remain poorly understood, and its potential as an indicator of immune competence is unknown, despite the convenience of non-invasive skin sampling. The expression of immune-related molecules in fish skin could significantly contribute to the immune response against infections. Some observed differences in cytokine genes in disease susceptibility between fish species and/or strains have been linked to the differing ability of the fish to prevent pathogen attachment and entry at mucosal epithelial sites (*Adamek et al., 2019*; *Adamek et al., 2022a*; *Adamek et al., 2022b*). The gill mucosal immune system is distinguished by various humoral and cellular immune mechanisms that synergize to safeguard the tissue against infection (*Gomez, Sunyer & Salinas, 2013*). When infected, both the local immune cell populations, which include mucosal "innate" T-cells and IgT + B cells, and the immune cells mobilized from specialized immune organs *via* the bloodstream, can contribute to the elimination of the infection (*Marcos-López et al., 2017*).

The primary constituents of the mucus layer are large, filamentous glycoproteins known as mucins that are highly glycosylated. They are strongly adhesive and play a crucial role in protecting the mucosal surfaces (*Roussel & Delmotte, 2005*). Mucins impart viscosity to mucus and form a framework within which various antimicrobial molecules are present (*McGuckin et al., 2011*). As mentioned earlier, excessive mucus production during CEV infection is one of the major clinical symptoms. Therefore, conducting experimental trials to understand the role of mucin in fish affected by CEV would be interesting.

It has been proven that gills are the known target tissue of CEV and are crucial for the biology of the all viral genogroups (*Adamek et al., 2017c*). Therefore, in the present study, the initial part consists of an evaluation of immune gene responses to CEV genogroup I and IIa infections in the gills of different carp strains (AS, koi, PS, and Rop). In addition, to determine whether or not mucosal immune genes fully respond to CEV infection, therefore, two carp strains (KSD-susceptible koi and KSD-resistant AS) were exposed to CEV genogroup IIa. In the experimental trial previously published by *Adamek et al. (2017c)*, the expression of type I interferon and interferon-stimulated genes encoding IFNa2 were reported in the gills of common carp strains. The background and a brief summary of CEV load and replication for the study groups were compiled from previous experimental trials and can be found in Tables S1 & S2. Here, we focus on the Co II and

Co IV infectious trials, which were infected with genogroup I and IIa, respectively, and from which the samples were analyzed. In the current study, to measure changes in gene expression of selected immune genes, reverse transcription-quantitative PCR (RT-qPCR) assays have been used. Our data provide a deeper understanding of CEV pathogenesis through cytokine and other immune-related genes.

## MATERIALS & METHODS
### Infections with CEV genogroup I and IIa
*Experimental samples acquired*

The samples analyzed in the present study are comprised of two experimental parts. According to the 3R rule, the samples obtained during experimental infections were reused for additional analysis of the immune responses. The experimental procedures were approved by Lower Saxony State Office for Consumer Protection and Food Safety (LAVES), Oldenburg, Germany under the reference number: 33.19–425 2-04-16/2144. In the initial part, the samples used for analysis originated from the previously published infection trial of CEV in different carp strains (*Adamek et al., 2017c*). Naïve recipient common carp strains Amur wild carp (AS), Ropsha scaly carp (Rop), Prerov scaly carp (PS), and koi were acquired as swimming fry from the University of South Bohemia in Ceske Budejovice, Faculty of Fisheries and Protection of Waters, located in Vodnany, Czech Republic. Using a full-factorial mating scheme of three females with three males, each experimental stock was established by artificial reproduction of the appropriate carp strain (*Kocour et al., 2005*). From the time the eggs were laid, the stocks were maintained in a closed recirculation system that was supplied with tap water. Later on, fry were transported to the facility of the University of Veterinary Medicine in Hannover and raised in a recirculation system at 20 °C in tap water. Commercial carp feed (Skretting, Norway) was fed to fish at 1% of body weight per day. The average weight of the fish at the beginning of the infection experiments was 3.7 ± 0.9 g were kept under virus and parasite-free conditions. Each carp population was confirmed to be free of DNA/RNA specific for CyHV-3, spring viremia of carp virus (SVCV), and CEV by the mean of RT-qPCR or qPCR before using them in infection experiments.

The infections of tested fish have been carried out by cohabitation with either CEV-affected common carp or koi showing clinical signs of the disease. Both CEV-affected common carp and koi were examined by the mean of end-point PCR (already published data) (*Adamek et al., 2017c*) which confirmed CEV genogroups I and IIa, respectively.

*Sample analysing theme for the experimental part one*

Total four cohabitation experiments (Co I, II, III, and IV) were performed with all four strains. More detailed information for all the infectious trials mentioned above can be found in *Adamek et al. (2017c)*. The present study was only focused on the samples from cohabitation experiment II (Co II) and IV (Co IV). In the Co II trial, 11 individuals from all four strains AS, koi, PS, and Rop were cohabited with CEV genogroup I affected common carp. On days 6 and 11 post-infection, four fish from each strain were euthanized by immersion into a 0.5 g L-1 tricaine (Sigma) solution. The samples from the gills were

collected individually in RNAlater for RNA isolation and gene expression analysis. During the Co IV experiment, the same four carp strains (eight fish per strain) were cohabited with koi infected with CEV genogroup IIa. Subsequently, four fish from each strain were euthanized with the above-mentioned method at days 6 and 11 post-infection. Their gills were collected in RNAlater. Non-infected fish from each strain were used as a negative control for gene expression in both experiments.

***Fish rearing and sample analysing theme for the second experiment part***

In the second experimental study, naïve AS and koi with an average body weight of 20.4 ± 10.9 g were used to evaluate the selected immune-related gene expressions in the mucosal organs to CEV infection. Before the start of the experimental infection trial, fish were kept in the similar condition mentioned above and confirmed to be free from DNA/RNA specific for CyHV-3, spring viremia of carp virus (SVCV) and CEV. Feeding was done using a commercial feed (Perla Plus, Skretting, Norway) at a rate of 1% body weight per day. In this part of the study, the data were collected as described by *Adamek et al. (2017c)*. Total of five fish from each strain were cohabitated with koi infected with CEV genogroup IIa. At day 5 post-infection, fish from both strains were euthanized by immersion into a 0.5 g L$^{-1}$ tricain (Sigma) solution, and mucosal tissues such as gill, gut and skin were collected in RNAlater. Fish of the same strains sampled in the day 0 were used as a non-infected control for gene expression analysis.

## DNA isolation

DNA was isolated as previously described in *Adamek et al. (2017c)*. Specifically: 25 mg of tissue was mechanically lysed in a QIAgen Tissuelyser II (Qiagen, Hilden, Germany), then the QIAamp DNA Mini Kit (Qiagen, Hilden, Germany) according to the manufacturer's instructions was used. After isolation, the samples were diluted to 50 ng µL $-1$ and stored at $-80\,°C$.

## RNA extraction and cDNA synthesis

Total RNA was extracted from the 25 mg of RNAlater-stored tissue samples using Tri-reagent (Sigma, St. Louis, MO, USA), according to the manufacturer's instructions. The remaining genomic DNA was digested with 2 U of DNase I according to the manufacturer's protocol. Prior to cDNA synthesis, RNA concentrations were determined by spectrophotometry, the integrity was checked using a 1.5% agarose gel. Complementary DNA (cDNA) was synthesized from 300 ng of total extracted RNA using Maxima TM First Strand cDNA Synthesis Kit (Thermo Fisher Scientific, Waltham, MA, USA). Prior to RT-qPCR analysis, cDNA samples were diluted 1:20 with nuclease-free water. Concentrations of the samples were determined so that homogeneous RNA could be produced for cDNA synthesis. cDNA samples were stored in $-20\,°C$ until further use.

## RT-qPCR/qPCR analysis

For the estimation of CEV load from CEV genogroup IIa, a qPCR-based double-labelled probe was used as described by *Adamek et al. (2017b)*. RT-qPCR assays were carried out using synthesized cDNA and specific primers listed in Table 1 to examine the expression of
all selected immune-associated genes in gills from part one of the experiment and gill, skin and gut from the second part of the experiment. The RT-qPCR reactions were carried out in duplicates using Maxima SYBR Green 2x master mix (Thermo Fisher Scientific, Waltham, MA, USA) in StepOne Plus Cycler (Thermo Fisher Scientific, Waltham, MA, USA). The reaction mixture was prepared as follows: $1 \times$ Maxima SYBR Green mastermix (with 10 nM of ROX), 0.2 µM of each primer, 3.0 µL of $20 \times$ diluted cDNA and nuclease-free water to a final volume of 10 µL. The following amplification program was used: initial denaturation (10 min at 95 °C) followed by 40 cycles of denaturation (30 s at 95 °C), annealing (30 s at 55 °C), and elongation (30 s at 72 °C). The non-template and minus reverse transcriptase (-RT) controls were performed for each reaction mix and cDNA sample, respectively. Obtained RT-qPCR data were analyzed using the StepOne software version 2.1 by measuring and analyzing the quantitative cycles (Cq) for every reaction and exported to Microsoft Excel. To determine the amount of particular gene copy numbers present in each sample, recombinant plasmid standard curve from $6 \times 10^0$ to $6 \times 10^6$ gene copies were prepared and used. To normalize expression, the 40S ribosomal protein S11 was used as a reference gene. Using the formula below, the level of gene expression was calculated as the copy number of the gene normalized against $1 \times 10^5$ copies of 40S ribosomal protein S11 (normalized copy number):

Normalized copy number = mRNA copies per PCR for target gene/(mRNA copies per PCR for reference gene/$10^5$).

## Statistical analysis

Initially, raw RT-qPCR data were analyzed using StepOne software v2.1. The normalized gene expression data were transformed using a Log 10 ($x$) transformation before statistical analysis. Statistical analysis was performed using SigmaPlot 12.5 software (Systat Software, Chicago, IL, USA). To detect significant differences ($p \leq 0.05$) in gene expression and viral load during CEV infection, 1-way or 2-way ANOVA were used along with pairwise multiple comparisons using the Holm-Sidak method.

## RESULTS

### CEV viral load and replication

The viral load and replication of the samples have been determined and published previously (*Adamek et al., 2017c*). The common carp strains infected with CEV affected carp with genogroup I manifested significant differences in susceptibility to the infection. Koi and PS were more susceptible than AS and showed a high viral load and CEV mRNA expression than other strains. For additional details, please refer to Table S1.

When the common carp strains and naïve koi were infected with CEV genogroup IIa (experiment Co IV), koi showed the highest viral load from all strains in both sampling times (151,668 mean copies at 6 dpi, 1,253,267 copies at 11 dpi), which confirmed its higher susceptibility to this genogroup. See Table S2 for more information.

**Table 1  RT-qPCR primers used in the study.**

| Target gene | Sequences | GeneBank ID or reference |
|---|---|---|
| IL-10 | CGCCAGCATAAAGAACTCGT | |
| | TGCCAAATACTGCTCGATGT | AB110780 |
| IL-1 $\beta$ | AAGGAGGCCAGTGGCTCTGT | |
| | CCTGAAGAAGAGGAGGCTGTCA | AJ245635 |
| IL-6a | CAGATAGCGGACGGAGGGGC | |
| | GCGGGTCTCTTCGTGTCTT | KC858890 |
| TNF- $\alpha$2 | CGGCACGAGGAGAAACCGAGC | |
| | CATCGTTGTGTCTGTTAGTAAGTTC | AJ311801 |
| Mx2 | ATGACCCAGCAGAAGTGGAG | |
| | CAGGAACATTGGCAGAGATG | XM_019081222 |
| IgM | CACAAGGCGGGAAATGAAGA | |
| | GGAGGCACTATATCAACAGCA | AB004105 |
| CD4 | CGTGGACATCTGGCTTTGTG | |
| | TTTGGTTTTGCGTCGTCTGT | DQ400124 |
| CD8b1 | CGGCTCGGAAACTATCACCT | |
| | GAGTGGCGGACAGGTTTTCTC | EU025120 |
| *GzmA* | GTGTTGGCATCGTCAGTTACG | |
| | AGTACCCCAACCTGTCACG | GU362096 |
| 40S | CCGTGGGTGACATCGTTACA | AB012087 |
| | TCAGGACATTGAACCTCACTGTCT | |
| Muc5b | CAGCCCTCTTCCTCTTTCATC | |
| | CCACTCATCTTTCCTTTCTCTTC | |
| Muc2c | TGACTGCCAAAGCCTCATTC | *Van der Marel et al. (2012)* |
| | CCATTGACTACGACCTGTTTCTC | |

## Expression of IL-10, IL-1 $\beta$, TNF-$\alpha$2 and IL-6a genes in carp strains during the infection with CEV genogroups I and/or IIa

In the challenge with CEV genogroup I (Co II), koi depicted significant lower level of IL-10 and higher levels of IL-6a on day 6th post-infection compared to day 0 p.i. On 11th dpi, koi showed significantly higher level of IL-1$\beta$, TNF-$\alpha$2 and IL-6a genes, and significant down-regulation of IL-10 gene expression compared to day 0 (Fig. 1). Whereas, in AS, the expression of IL-10 was at the same level on both days 6 and 11 p.i. compared to day 0. Significant up-regulation of IL-1$\beta$, and IL-6a genes were noticed in AS only on day 11 p.i. Significant elevated levels TNF-$\alpha$2 were found on both days 6 and 11 p.i. In PS strain, expression of IL-1$\beta$ was significantly up-regulated on day 6 p.i. However, no significant differences were detected in the expression of other immune genes such as IL-1$\beta$, TNF-$\alpha$2 and IL-6a on both time points compared to fish sampled on day 0. In Rop, significant up-regulation of TNF-$\alpha$2 and IL-6a was observed on both days 6 and 11 p.i. However, there was no differences in the expression of IL-10, and IL-1$\beta$ on both sampling days.

During the course of an infection with CEV genogroup IIa (Co IV), koi evinced significant up-regulation of IL-10 and IL-1$\beta$ on both days 6 and 11 p.i. compared to day

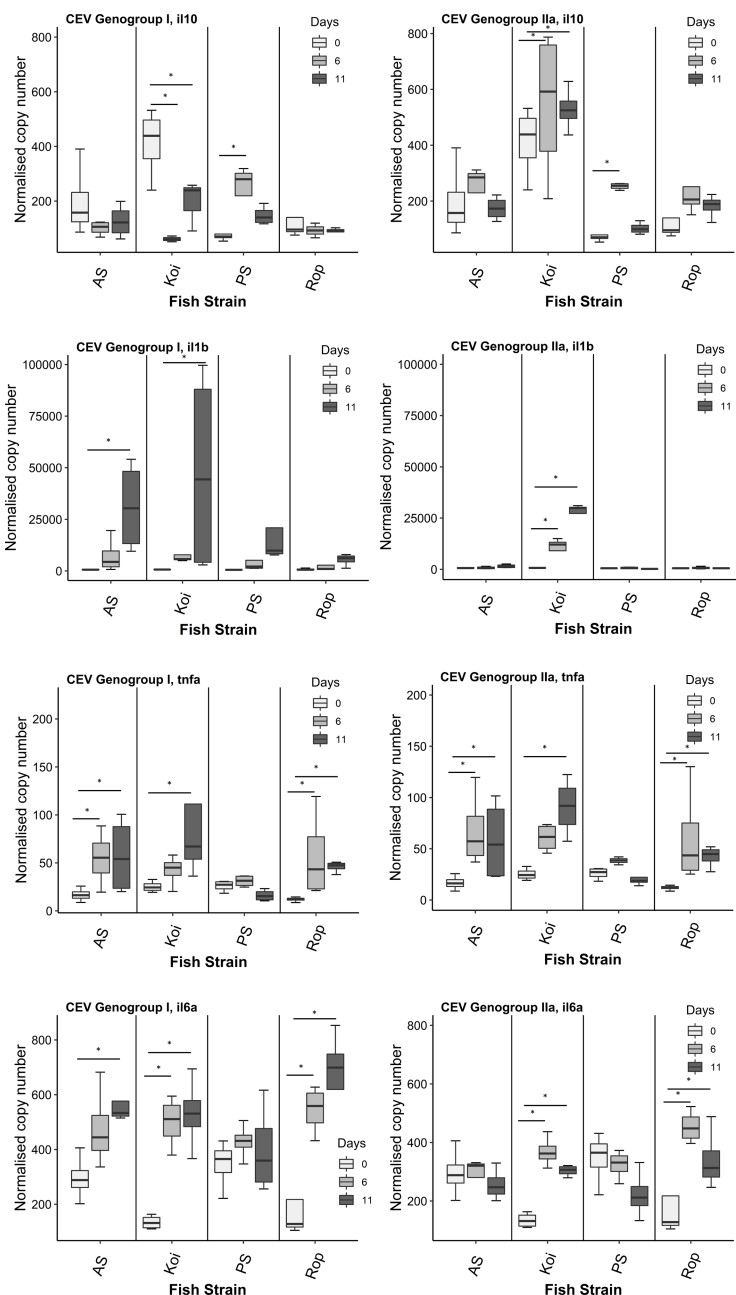

**Figure 1  Reverse transcriptase quantitative PCR (RT-qPCR) analysis of IL-10, IL-1 β, TNF- α2 and IL-6a genes in gills of four strains.** Gene expression was measured in the gill of carp individuals from different strains (AS, koi, PS, and Rop) at days 0 (non-infected), 6 and 11 post-infections of CEV genogroup I or IIa. The gene expression was normalized to the expression of the gene encoding the S11 protein of the 40S subunit as a reference gene. Data are shown as box plot indicating mean and standard deviation from $n = 4$ fish. Asterisks denote statistically significant differences ($*p < 0.05$) between the control (day 0) and infected once (day 6 and 11).

0. On day 6 p.i., TNF-$\alpha$2 levels was not significantly higher on 6 dpi and the increase continued to 11 dpi, when it was significantly higher than control (0 dpi). The expression level of IL-6a was significantly higher on day 6 p.i., then it gradually decreased but remained significantly up-regulated also on day 11 p.i. comparing with day 0. In AS, the expression of IL-10 gene was at the same level on both days 6 and 11 p.i. compared to day 0. No significant differences were observed also in IL-1$\beta$ levels on both days 6 and 11 p.i. Significantly higher level of TNF-$\alpha$2 was noticed on both days 6 and 11 p.i., and these levels were much higher when compared to genogroup I infection in AS. In PS strain, only expression of IL-1$\beta$ was significantly up-regulated on day 6 p.i. compared to day 0. No significant differences were detected in the expression of other immune genes such as IL-1$\beta$, TNF-$\alpha$2 and IL-6a on both time points compared to fish sampled on day 0. In Rop, significant up-regulation of TNF-$\alpha$2 gene on both days. The expression of IL-6a gene was also up-regulated on days 6 and 11 p.i. with slight decrease between 6 and 11 dpi. Furthermore, there was no differences in the expression of IL-10, and IL-1$\beta$ on both days 6 and 11 p.i.

## Expression of CD4, CD8b1 and GzmA in carp strains during the infection with CEV genogroups I and/or IIa

Challenge with CEV genogroup I (Co II), significantly down-regulated the expression of CD4 and *GzmA* in koi on days 6 and 11 p.i. compared to day 0 (Fig. 2). We did not observe any differences in the expression of CD8b1 in koi on both days 6 and 11 p.i. AS showed remarkable significant down-expression of CD8b1 and *GzmA* genes on both infected days 6 and 11 compared to day 0. In addition, the expression level of CD4 was also significantly down-regulated on days 6 and 11 p.i. but the response was not stronger as CD8b1 and *GzmA* genes when compared to day 0. Furthermore, PS and Rop strains did not show any significant differences in the expression of CD4, CD8b1 and *GzmA* on day 6 and 11 p.i. compared to fish sampled on day 0.

In the challenge with CEV genogroup IIa (Co IV), koi manifested significant down-regulation of CD4 on day 6 and 11 p.i. compared to day 0. However, there was no CD8b1 response detected in koi on both infected days compared to control group. The gradual decrease of *GzmA* was noted in koi on both day 6 and 11 p.i compared to day 0, but with significant difference on day 11 only. There was no difference in CD4 expression in AS strain p.i compared to day 0. However, significant down-regulation of CD8b1 gene expression was noticed in AS on both days 6 and 11 p.i when compared to day 0. Furthermore, slight down-regulation of *GzmA* was seen but without significant difference. In PS and Rop no significant differences were observed in the expression of CD4, CD8b1 and *GzmA* on both time points compared to fish sampled on day 0.

## Expression of Mx2, IgM, and Mucin genes in carp strains during the infection with CEV genogroups I and/or IIa

During the challenge with CEV genogroup I (Co II), the significantly higher expression level of Mx2 was noticed in koi on day 6 p.i. however on day 11 p.i. the levels were restored near day 0 (Fig. 3). The expression of IgM in koi was significantly down-regulated on day 11 p.i. compared to day 0. On day 6 p.i., IgM levels was also lower but not significant when compared to control group. In AS, on both days 6 and 11 p.i. we did not observe

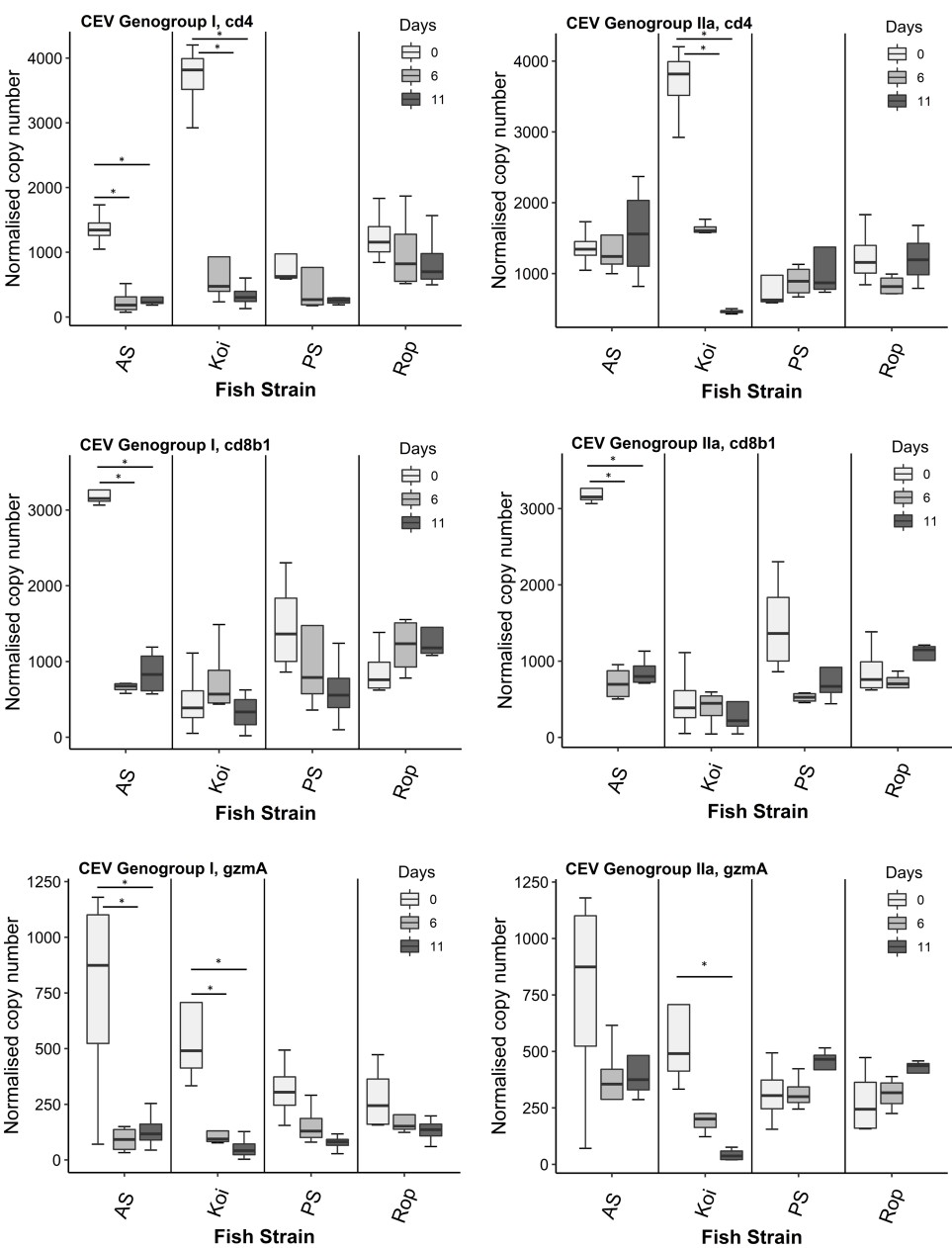

**Figure 2** **Reverse transcriptase quantitative PCR (RT-qPCR) analysis of CD4, CD8b1 and *GzmA* genes in gills of four strains.** Gene expression was measured in the gill of carp individuals from different strains (AS, koi, PS, and Rop) at days 0 (non-infected), 6 and 11 post-infections to CEV genogroup I and IIa. The gene expression was normalized to the expression of the gene encoding the S11 protein of the 40S subunit as a reference gene. Data are shown as box plot indicating mean and standard deviation from $n = 4$ fish. Asterisks denote statistically significant differences (*$p < 0.05$) between the control (day 0) and infected once (day 6 and 11).

any significant difference in the expression of Mx2 gene compared to day 0. The IgM levels in AS was significantly down-expressed on day 6 p.i. compared to control fish. Mx2 expression in PS strain was significantly peaked on day 6 p.i. and stayed elevated up to day

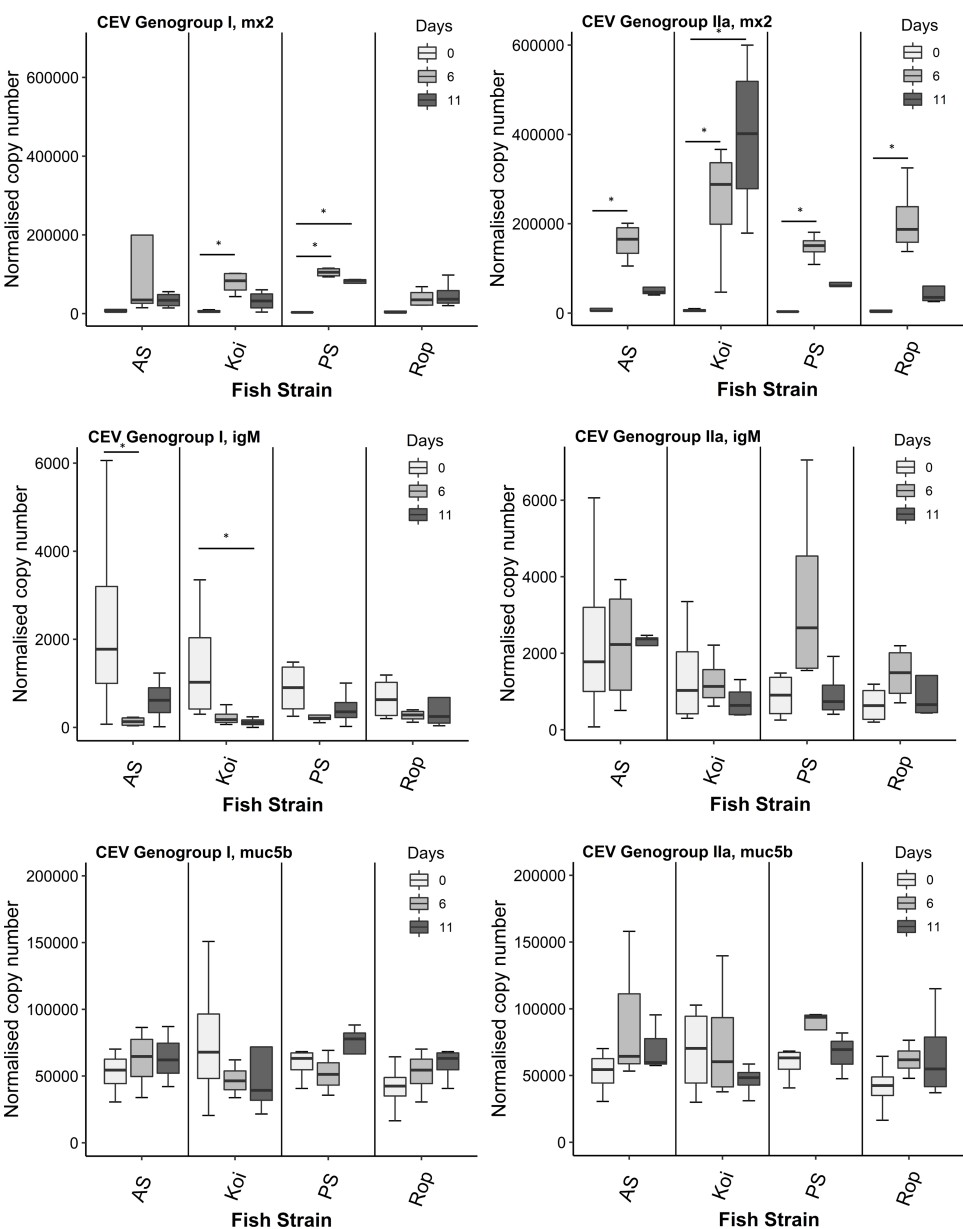

**Figure 3** **Reverse transcriptase quantitative PCR (RT-qPCR) analysis of Mx2, IgM, and Mucb5 in gills of four strains.** Gene expression was measured in the gill of infected and non-infected (day 0) carp individuals from different strains (AS, koi, PS, and Rop) at days 6 and 11 post-infection to CEV genogroup I and IIa. The gene expression was normalized to the expression of the gene encoding the S11 protein of the 40S subunit as a reference gene. Data are shown as box plot indicating mean and standard deviation from $n = 4$ fish. Asterisks denote statistically significant differences (*$p < 0.05$) between the control (day 0) and infected once (day 6 and 11).

11 p.i. compared to control group. No IgM response was detected in PS strain on both infected days. Rop did not exhibit a significant shift in expression level of Mx2 or IgM on either day 6 or day 11 p.i compared it to control.

**Table 2 CEV virus load in tissues from AS and Koi after experimental cohabitation with CEV genogroup IIa.**

| | AS<br>Day 5 | Koi<br>Day 5 |
|---|---|---|
| Gill | | |
| Mean | 2.09E + 04[a] | 2.51E + 05[bc] |
| Median | 7.86E + 03 | 2.52E + 05 |
| SD | 3.09E + 04 | 9.68E + 04 |
| Gut | | |
| Mean | 1.08E + 02[c] | 1.83E + 03[c] |
| Median | 3.20E + 01 | 1.34E + 03 |
| SD | 1.96E + 02 | 2.05E + 03 |
| Skin | | |
| Mean | 5.29E + 02[c] | 1.10E + 05[b] |
| Median | 1.21E + 02 | 7.50E + 04 |
| SD | 1.08E + 03 | 1.10E + 05 |

**Notes.**

Carp edema virus load was measured by qPCR as copy numbers of virus specific DNA in the gill, gut and skin of AS and koi during CEV genogroup IIa infection. Samples were collected 5-day post-exposure from $n = 6$ fish per day. The data on virus load is shown as mean, median and standard deviation (SD) of genome copies in 250 ng of isolated DNA. Different letters indicate significant differences at $p \leq 0.05$ between carp strains.

Upon challenged with CEV genogroup IIa (Co IV), in koi there was observed a significant upward trend of Mx2 gene on both days 6 and 11 p.i. in comparison to the control day 0. The IgM expression in koi gradually slightly decreased but without statistical significance. In AS, the significant peak level of Mx2 was noticed on day 6 p.i. and reduced on day 11 p.i. near to the control group. There were no differences found in IgM transcripts in AS at both infected days compared to control fish. Both PS and Rop strains significantly up-regulated Mx2 expression levels on day 6 p.i. and then reduced to the level of control day 0. We did not observe any differences in IgM levels in both PS and Rop strains on both days 6 and 11 p.i. compared to the fish sampled on day 0. The present study also investigated the effects of CEV genogroup I or IIa exposure on Muc5b gene expression in the gills of four different fish strains. Results demonstrated that there was no significant difference in the Muc5b gene expression on both days 6 and 11 p.i. in the gills of all four strains when compared to the control. However, a non-significant slight decrease in the Muc5b mRNA level on day 11 was noted in the gills of koi carp following the exposure to CEV genogroup I or IIa, which differed slightly from the other fish strains.

## Virus load in AS and koi carp infected with CEV genogroup IIa

The virus load was analyzed in the gill, gut and skin tissues at day 5 post-infection. Among all selected organs, gills harboured the highest number of CEV specific DNA copies with a mean of 251,130 copies and median of 252,444 in koi carp and mean of 20,934 and median 7,859 of copy numbers in AS per 250 ng of extracted DNA (Table 2). The skin of koi having the second highest viral load with a mean copy number of 110,457, and median 74,986, whereas the gut and skin organs from AS had a virus load below 1,000 copies per 250 ng of isolated DNA.

### Expression of immune-related genes in the gill, gut and skin tissues of AS and koi during the infection with CEV genogroup IIa

### IL-10, IL-1$\beta$, TNF-$\alpha$2 and IL-6a genes

The expression level of all selected cytokine genes such as IL-10, IL-1$\beta$, TNF-$\alpha$2 and IL-6a were significantly up-regulated in the gill and gut tissues of koi carp at day 5 p.i. compared to day 0 (Fig. 4). In contrast, the expression of none of the cytokine genes was shifted in both gill and gut tissue of AS. Only IL-10 gene showed slight lower expression in the gill on day 5 p.i. compared to day 0, but without a significant difference.

In the skin tissue, no significant changes were observed in cytokine genes expression in both koi and AS strains at day 5 p.i. compared to the fish sampled on day 0, except of IL-1$\beta$ gene, which was significantly up-expressed ($p < 0.01$) at day 5 p.i. compared to day 0.

### CD4, CD8b1 and GzmA genes

There was a significant down-regulation ($p < 0.05$) of CD4 and *GzmA* transcript levels observed in the gill of koi carp at day 5 p.i. compared to the control group (Fig. 5). However, no CD8b1 response was detected in koi at day 5 p.i., compared to the control day 0. In the AS strains' gill, slight down-regulation was observed in CD4, CD8b1, and *GzmA* levels on day 5 p.i. compared to day 0, although the levels were not statistically significant.

In the gut and skin of both carp strains, no significant changes were noticed in CD4, CD8b1, and *GzmA* transcripts on day 5 p.i. compared to the fish sample on day 0.

### Mx2, IgM and Mucin genes

No significant difference in IgM levels were noticed in the gill and gut in both AS and koi strains on 5 dpi, compared to the control fish. Significant down-regulation of IgM was noticed in koi's skin at day 5 p.i. however, slight down-expression of IgM transcripts was found in the AS skin and gill tissue but without significant changes (Fig. 6).

The expression of the Mx2 gene in gill, gut and skin tissues in AS strain was significantly up-regulated on day 5 post-CEV exposure compared to day 0. The Mx2 transcripts in koi gill and gut were also significantly up-expressed on day 5 p.i. even much higher than AS strain. Further, koi skin tissue did not show any significant change on day 5 p.i. compared to the control.

The tissue-specific expression of mucin genes, Muc2 and Muc5b, was observed among gill, gut, and skin. Muc2 was exclusively expressed in the gut tissue, whereas Muc5b was expressed in the gill and skin tissues. No significant differences were found in the expression levels of these genes in gills and gut. However, a significant down-regulation of Muc5b transcripts, was observed in the skin tissues of koi on day 5 post-infection compared to the control.

## DISCUSSION

### Immune gene expressions during the infections with CEV genogroup I and IIa

The differences in the immune responses occurring between the fish strains with different susceptibility to pathogen could explain how the fish successfully protect themselves from

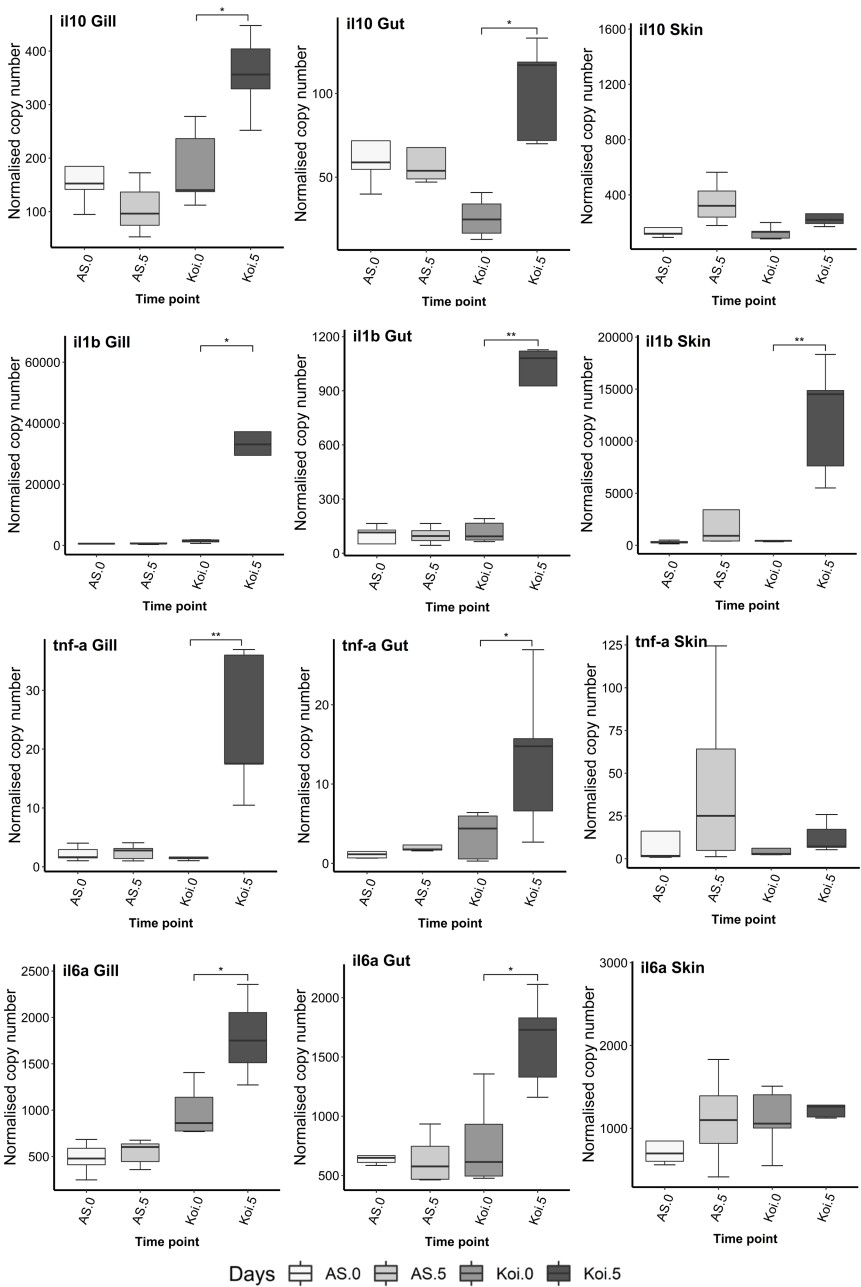

**Figure 4** Reverse transcriptase quantitative PCR (RT-qPCR) analysis of IL-10, IL-1 $\beta$, TNF- $\alpha$2 and IL-6a genes in the gill, gut and skin tissues of AS and koi at day 0 (non-infected) and 5 post-infection of CEV genogroup IIa. The gene expression was normalized to the expression of the gene encoding the S11 protein of the 40S subunit as a reference gene. Asterisks denote statistically significant differences marked with * at $p < 0.05$, with **$p < 0.01$ between the control and infected once.

the infections. However, for fish poxviruses there is underwhelming scarcity of data. The immune responses were addressed only in handful of works; with only type I interferon response characterized during CEV infection in more than two carp strains (*Adamek et al.*,

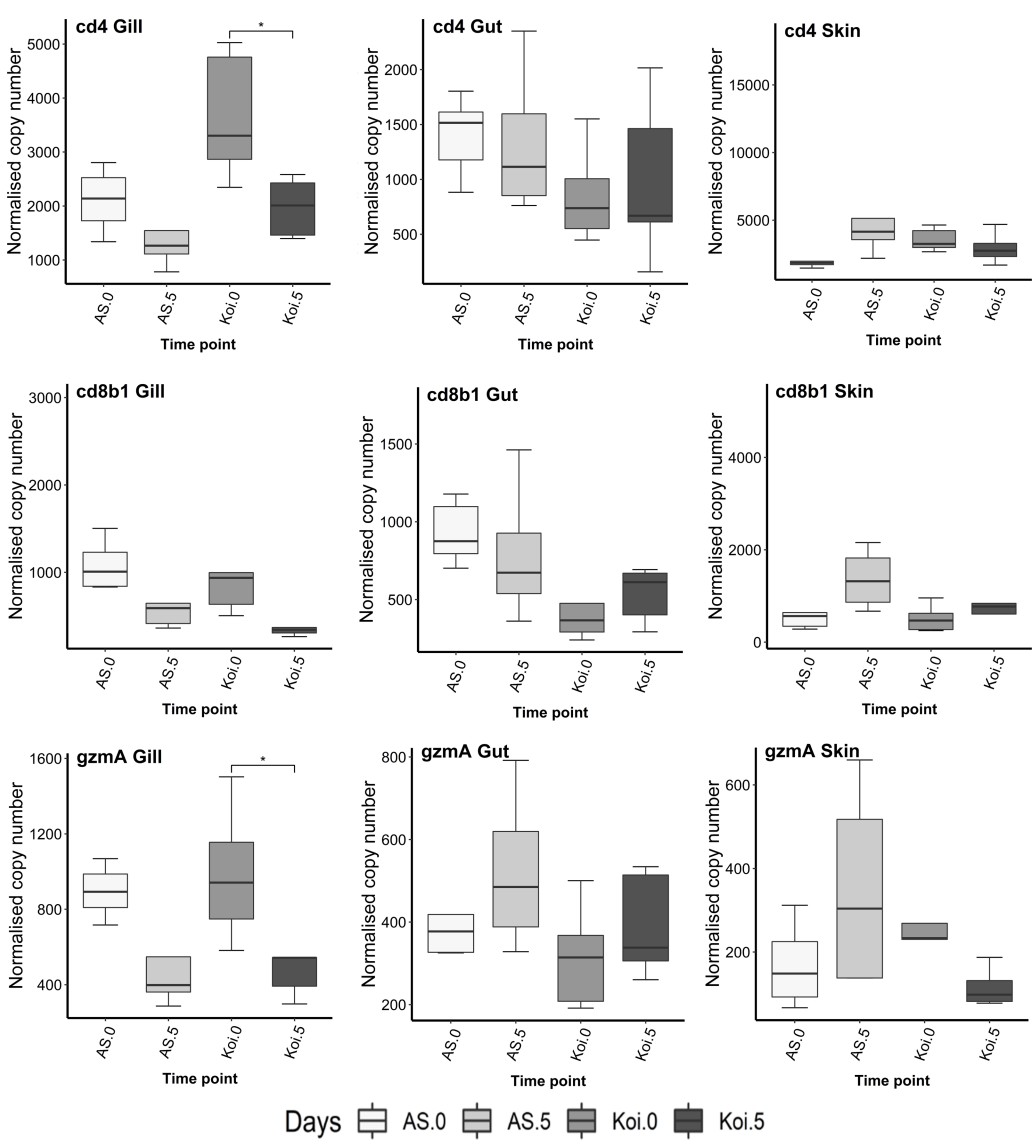

**Figure 5** Reverse transcriptase quantitative PCR (RT-qPCR) analysis of CD4, CD8b1 and *GzmA* in the gill, gut and skin tissues of AS and koi at day 0 (non-infected) and 5 post-infection of CEV genogroup IIa. The gene expression was normalized to the expression of the gene encoding the S11 protein of the 40S subunit as a reference gene. Asterisks denote statistically significant differences marked with * at $p < 0.05$, with **$p < 0.01$ between the control and infected once.

*2017c*). Furthermore, no studies describing the any other immune responses of carp strains after infection by cohabitation with fish infected with the CEV genogroup I or IIa have been reported yet. Therefore, our study is the first to report the looking on inflammation and adaptive immune responses in carp strains during CEV infections.

Cytokines belong to the most widely studied group of molecules involved in the function of the immune system. Amongst, a key pro-inflammatory cytokine, interleukin-1$\beta$ (IL-1$\beta$) plays an essential role in the fish immune system (*Zou & Secombes, 2016*). According to the

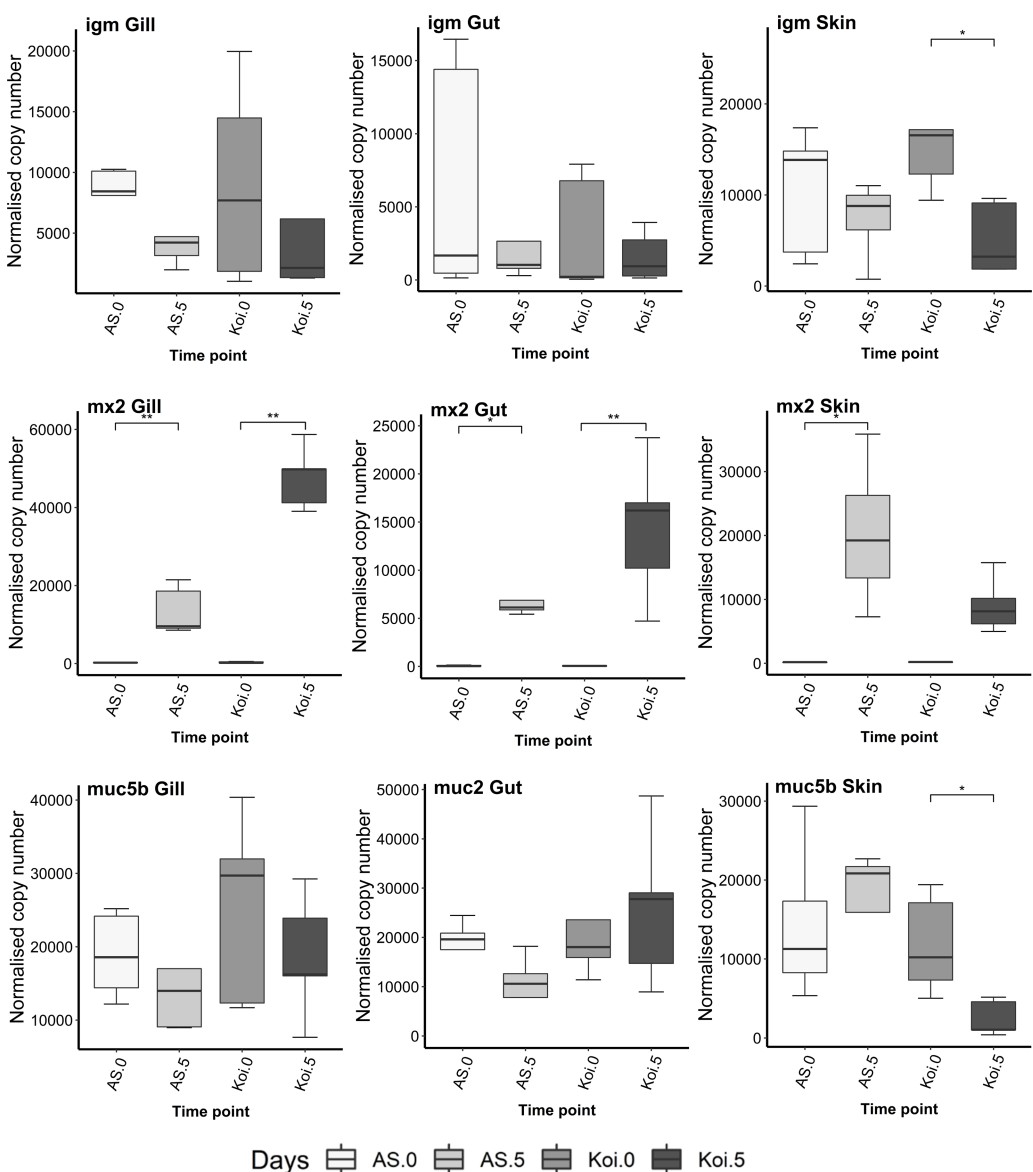

**Figure 6** Reverse transcriptase quantitative PCR (RT-qPCR) analysis of Mx2, IgM, Muc2, and Mucb5 in the gill, gut and skin tissues of AS and koi. RT-qPCR analysis of Mx2, IgM, Muc2, and Mucb5 in the gill, gut and skin tissues of AS and koi at day 0 (non-infected) and 5 post-infection of CEV genogroup IIa. The gene expression was normalized to the expression of the gene encoding the S11 protein of the 40S subunit as a reference gene. Asterisks denote statistically significant differences marked with * at $p < 0.05$, with **$p < 0.01$ between the control and infected once.

previous studies, it has been studied in a wide range of fish species (*Buonocore et al., 2003*; *Wang et al., 2006*). In our study, during both experiments (Genogroup I and Genogroup IIa infections), only koi carp showed up-regulation of IL-1$\beta$ post-infection. This strong IL-1$\beta$ inflammatory response may coincide with the previously conducted study where koi were found to be susceptible to both genogroups I or IIa infections (*Adamek et al., 2017c*). Furthermore, high expression of IL-1$\beta$ was also found in different carp lines at

days 3 and 5 post-infection challenged with CyHV-3 (*Rakus et al., 2012*). IL-1$\beta$ might be the significant driver of the gill pathology developing during the infection with CEV. This molecule was associated with the increased pathological changes in mucosal tissues during CyHV-3 vaccination studies (*Adamek et al., 2022a*) as well as tilapia lake virus (TiLV) susceptible strains of Nile tilapia (*Oreochromis niloticus*) (*Adamek et al., 2022b*). In both non-vaccinated carp which was subsequently challenged with CyHV-3 and TiLV susceptible strains of tilapia infected with the TiLV, the disruption of the skin and gill barriers and increased occlusion of the intralaminar spaces was associated with increased expression of the gene encoding for IL-1$\beta$. Current studies showing the increased IL-1$\beta$ expression in the CEVD susceptible strains could explain the pathology recorded in these fish (*Adamek et al., 2017c*).

In our case, the low level of IL-1$\beta$ in other strains might be due to the increased expression of IL-6a, which is a key immunosuppressive cytokine secreted by regulatory T-cells (*Zou & Secombes, 2016*). This is in accordance with the previous study where IL-6a significantly down-regulated the expression of IL-1$\beta$ and TNF-$\alpha$ (key pro-inflammatory cytokines), suggesting its potential anti-inflammatory role in trout (*Costa et al., 2011*). In the current study, significant changes in the expression level of IL-6a were observed in AS, koi, and Rop strains infected with CEV genogroup I or IIa. Comparatively, the expression pattern was different between the genogroup I and IIa. Fish infected with genogroup I evidenced higher expression on days 6 and 11 compared to the control group. However, IL-6a was elevated on day 6 post-infection and declined on day 11 but remained higher than in the control. In common carp, the role of IL-6a in the response to viral infections is not defined yet. However, its crucial role in the humoral immune response was well observed in other fish species, such as Nile tilapia challenged with the bacterial infection which promotes IgM antibodies production (*Wei et al., 2018*). In fish, the expression of the IL-10 gene has also been linked to a decrease in inflammatory responses (*Forlenza, 2009*; *Ingerslev et al., 2009*; *Raida & Buchmann, 2008*). We observed relatively high down-regulation of the expression of IL-10 compared to the control group during the genogroup I infection on day 6 in koi carp. This low level was maintained till day 11 post-infection. On the contrary, the koi infected with genogroup IIa elevated the IL-10 expression on days 6 and 11 post-infection. IL-10 up-regulation to genogroup IIa could result from an increase in humoral immunity and inhibition of inflammation. Tumor necrosis factor (TNF) is a critical cytokine that plays an essential role in physiological and pathological processes. It promotes phagocytosis and nitric oxide production in teleost during viral and bacterial infections (*Tafalla, Figueras & Novoa, 2001*). Our results revealed higher expression of TNF-$\alpha$2 in AS and Rop strains during both genogroups I and IIa infections. However, in koi, the significantly elevated level of TNF-$\alpha$2 was only found on day 11 post-infection campared to the control group. The higher levels of TNF-$\alpha$2 might be due to the elevation of IL-1$\beta$ expression. This can be consistent with the previous study where IL-1$\beta$ produced local effect on the expression of TNF-$\alpha$ in muscles that have been treated with a plasmid encoding IL-1$\beta$ in japanese flounder *Paralichthys olivaceus* (*Taechavasonyoo, Hirono & Kondo, 2013*).

In the context of poxvirus infections, T helper cells play a critical role in the activation and differentiation of other immune cells, such as B cells and cytotoxic T cells (CTLs)

(*Yamaguchi et al., 2019*). CD4 and CD8 are proteins or cell surface markers found on the surface of T cells, a type of white blood cell that plays a critical role in the immune response (*Zamoyska, 1998*). CD4 is primarily found on T helper cells, which are a type of T cell that play an important role in coordinating the immune response by secreting cytokines and helping to activate other immune cells such as B cells and cytotoxic T cells (CTLs) (*Brown, Román & Swain, 2004*). Changes in the gene expression of CD4 and CD8 markers can provide important information about the state of the immune system and its response to infection or disease. For example, a decrease in CD4 expression may indicate a decrease in T helper cell function, which can impair the immune response and increase susceptibility to infections (*Adamek et al., 2021*). In mammals, the down-regulation of CD4 is the anchor that disarms actions against the immunity of several poxviruses. In our study, down-regulation of CD4 was observed in AS and koi strains during both genogroup I or IIa infections at days 6 and 11 post-infection compared to the control group. The highest down-regulation was observed in koi carp on days 6 and 11 post-infection to both genogroups. Similar, down-regulation of the gene encoding for the CD4 receptor of T-cells in koi under CEV infection was detected in gills on days 6 and 9 post-infection (*Adamek et al., 2021*). They suggest that possible reason for CD4 down-regulation in their study could be due to the immunosuppressive effect, which results from hyperammonemia in infected CEV carp. Since, in our findings, a strong down-regulation of CD4 in koi carp could be speculated to be caused by the elevated viral transcripts infected with genogroup I or IIa post-infection. In addition, we further analyzed the expression of CD8b1 along with *GzmA* (a protein-coding lysis gene produced in cytotoxic T-cells) to CEV infections in all four strains. CD8 is primarily found on cytotoxic T cells, which are responsible for recognizing and killing virus-infected cells (*Mosmann, Li & Sad, 1997*). According to the study of *Adamek et al. (2021)*, down-regulation expression of CD8b1 was noticed in the gills of CEV-infected koi carp compared to infected AS at any time points (day 0 to 13 post-infection). Interestingly, in our findings, among all groups only AS showed a tendency of down-regulation from day 6 to 11 in genogroups I or IIa infections compared to the control groups. In contrary, when carp lines were challenged with CyHV-3, up-regulation of CD8b1 was noted from day 1 to day 5 post-infection (*Rakus et al., 2012*). Poxviruses have been demonstrated to modulate the host immune system through various mechanisms. These mechanisms include the inhibition of immune cell activation and proliferation, downregulation of immune gene expression, and production of immunomodulatory proteins (*Howard et al., 1998*). For example, some poxviruses are able to suppress the expression of interferons, which play a critical role in initiating the host antiviral response (*Fensterl & Sen, 2009*). These viruses can also hinder the activation of natural killer cells and T cells, two essential components of the host immune system. Based on these findings, it is plausible to suggest that the suppressed expression of CD4 and CD8b1 in fish infected with carp edema virus may be also a result of viral immunomodulation of the host immune system. The *GzmA* levels were strongly down-regulated in koi at days 6 and 11 post-infection compared to the control to both CEV genogroups. The down-regulation of *GzmA* in AS infected with genogroup I was observed on days 6 and 11 post-infection. This down-regulation expression of *GzmA* could be due to the lower recruitment of

cytotoxic cells in gills due to CEV infection, which can also be seen in the case of CD4 and CD8b1 expression. Interestingly, down-regulation of cytotoxic cell markers such as *GzmA* in gills has been associated with cortisol-induced immunosuppression during salmon gill poxvirus (SGPV) infection (*Amundsen et al., 2021*). Based on SGPV and current CEV results, down-regulation of *GzmA* appears to be a very important marker that could predict the development of the acute form of the disease caused by fish poxviruses (*Amundsen et al., 2021*).

Several hundred genes are induced by type I interferon (IFN I), including some that encode direct antiviral effectors, such as the Mx proteins (*Fernández-Trujillo et al., 2015*). They can impede viral replication at different stages of the virus's life cycle (*Das et al., 2019*). The innate immune responses mediated by type I interferon against viruses and bacteria have been demonstrated in fish (*Langevin et al., 2013*). The crucial role of type I interferon-inducing genes in the fish immune response has been emphasized, including those that encode direct antiviral effectors (*Machat et al., 2021*). Moreover, interferon-induced genes are known to encode interferon-stimulated proteins, such as myxovirus resistance and protein kinase R, which exhibit direct antiviral activity. These proteins can effectively inhibit viral transcription, degrade viral RNA, inhibit translation, or modify the proteasome to control various stages of viral replication (*Sadler & Williams, 2008*). It was shown that different strains of common carp exhibit varying levels of IFN and ISG expression in response to viral infections, with susceptible fish showing a quicker response than resistant ones (*Machat et al., 2021*). As stated earlier, interferons (IFNs) and IFN-stimulated genes provide the first line of defense against viral infections. However, viruses have evolved strategies to escape immune surveillance and establish successful infections (*Rai et al., 2021*). Therefore, it is critical to understand the complex mechanisms of the interaction between viruses and the host's innate immune system, particularly the role of IFNs and IFN-stimulated genes, in developing effective treatment strategies for acute viral infections

The induction of type 1 interferon responses, the mRNA expression of the genes encoding interferon alpha-2 and interferon-induced proteins viperin and RNA dependent protein kinase, have been determined in the previously conducted infection trials from the same experimental study (*Adamek et al., 2017c*). Furthermore, in a very recent study conducted by *Adamek et al. (2021)*, koi and AS under CEV infections displayed up-regulation of type I IFN (*ifnα2*) expression in the gill and kidney compared to infected fish at any time points (day 0 to 13 post-infection).

Another important interferon regulatory protein, known as myxovirus resistance protein (Mx), mediate cellular resistance against wide range of viral pathogens (*Gao et al., 2011*). Therefore, we further confirmed the antiviral response of Mx2 during CEV genogroup I and IIa infections. When the carp strains were infected with CEV genogroup I, the mRNA expression of the Mx2 gene was significantly up-regulated in the gills of koi, PS and Rop at both time points days 6 and 11 p.i. compared to day 0. However, the peaked level of Mx2 was observed on day 6 in koi and PS strains, which proved to be susceptible to CEV genogroup I (*Adamek et al., 2017c*). The expression level of Mx2 was not observed in AS strain at any time points when compared to day 0. During the course of an infection with CEV

genogroup IIa, the mRNA expression of Mx2 was highly significantly up-regulated in koi at 6 and 11 days post-infection compared to control. Carp from other strains also showed the up-regulation of mRNA encoding Mx2 protein (particularly at day 6 post-infection), but the magnitude of up-regulation was significantly lower on day 11 post-infection. The evidence from the previous studies (*Adamek et al., 2014*; *Adamek et al., 2017c*) shows that the antiviral response seemed positively correlated with the viral load in infected fish and cannot be related to the resistance of carp strains to infection. In our findings, the same results were established in the case of Mx2 antiviral response, except of in PS strain, where Mx2 levels were not correlated to viral load and replication. Our findings, regarding the antiviral response of Mx2 correspond with above mentioned results where type I IFNs and interferon-induced proteins were significantly up-regulated in carp strains evaluated by *Adamek et al. (2017c)* and *Adamek et al. (2021)*.

In a primary antibody response, it is the IgM antibodies that constitute the majority of the body's antibodies (*Janeway Jr et al., 2001*). Unlike other types of antibodies, IgM are produced primarily by B1 cells, with no apparent stimulation by specific antigens (*Tumang et al., 2005*). Antigen-specific IgM is produced early following infection by most pathogens, followed by IgA, IgG, and IgE antibody responses (such as IgT/Z and IgD in fish) that are more specific (*Boes, 2000*). Like koi herpes virus disease (KHVD), the first CEVD mortality in carp occurs approximately a week after infection at a water temperature of 18–24 °C, and mortality may reach 100% at 11 days following infection (*Miyazaki, Isshiki & Katsuyuki, 2005*; *Piačková et al., 2013*; *Lewisch et al., 2015*). This indicates that the mortality occurs prior the beginning of antibody production or at a time when the level of antibodies is too low allowing the virus to replicate more freely in the carp body. What is more important, our study indicate significant down-regulation of IgM levels noticed in koi on day 11 post-infection to CEV genogroups I or IIa compared to the control group which is an accordance with the findings of *Adamek et al. (2021)*, who reported the down-regulation expression of IgM in the gills of CEV infected koi with two-fold at day 3 and five-fold at day 9 p.i.. This could suggest that CEV infection process is delaying the start of antibody response however this requires further studies. Especially due to the fact, there were only slight variations in IgM levels in other carp strains but not significantly different.

The significant level of mucin conservation across all vertebrates serves to underscore their critical role in defending against pathogen intrusion. The pivotal function of mucin in the mucosal barrier of fish was demonstrated by the linkage of single nucleotide polymorphisms in mucin 2 and 5b with the resistance of rohu carp (*Labeo rohita*) to Aeromonas hydrophila infection (*Robinson et al., 2014*), thus underscoring the significance of mucin in host-pathogen interactions. In common carp, mucin 2 and mucin 5b expression was found to increase during beta glucan feeding (*Van der Marel et al., 2012*), while it decreased during CyHV-3 infection (*Adamek et al., 2013*), highlighting the versatility of mucin as a biomarker in different physiological settings. The present study investigated the impact of CEV genogroup I or IIa exposure on Muc5b gene expression in the gills of four distinct fish strains. The findings indicated that there was no substantial difference in the Muc5b gene expression on days 6 and 11 p.i. in the gills of all four strains compared to the control. Nonetheless, in the gills of koi carp, a negligible and non-significant decrease

in the Muc5b mRNA level was observed on day 11 after the exposure to CEV genogroup I or IIa, which was marginally different from the other fish strains. Our study differed from a previous investigation on common carp, which revealed that MUC5 gene expression in the fish gills was significantly up-regulated following dietary beta-glucan administration (*Van der Marel et al., 2012*). In zebrafish (*Danio rerio*), Similarly, administration of pectin in the diet resulted in a significant up-regulation of whole body MUC5 gene expression, as reported in a recent study (*Edirisinghe et al., 2019*). To date, the expression of other mucins in CEV-infected fish has only been investigated by *Adamek et al. (2017a)*. Their study reported a decrease in muc2-like transcripts in common carp at 144 h post-infection. In contrast, our study did not observe any significant changes in muc5b expression levels in CEV-infected carp gills at both 6 and 11 days post-infection. Hence, the potential role of muc5b in the susceptibility of CEV-infected strains to both genogroups of the virus remains unclear.

## Immune gene responses in mucosal tissues to CEV infection

In fish, the thymus and head kidney serve as the primary lymphoid organs, whereas the spleen, trunk kidney, and mucosa-associated lymphoid tissue (MALT) located in areas like the skin, gills, intestine, oral and nasal mucosa, and urogenital tract comprise the secondary lymphoid organs (*Ángeles Esteban, 2012*; *Soulliere & Dixon, 2017*). The MALT can be further divided by anatomical location into skin-associated lymphoid tissue (SALT), gut-associated lymphoid tissue (GALT), and gill-associated lymphoid tissue (GIALT) (*Ángeles Esteban, 2012*; *Salinas, Zhang & Sunyer, 2011*).

Skin, gills, and intestines of fish are the first barriers with the biggest mucosal surface providing the interface between a fish and its environment. As a defence mechanism against invading pathogens, these tissues secrete antimicrobial humoral factors, which act through various mechanisms to limit the spread and proliferation of pathogens (*Mehana, Rahmani & Aly, 2015*). In this study, AS and koi carps were exposed to CEV genogroup IIa to determine the immune gene expression in these mucosal tissues. Interestingly, the results indicate that not only the gills as a target tissue for the virus but also other mucosa react to the infection. Namely, the mRNA expression level of selected cytokine genes were mainly regulated in koi carp's gill and gut post-CEV infection. The detection of changes in expression of innate immune genes, such as IL-10, IL-1$\beta$, TNF-$\alpha$2 and IL-6a, during CEV infection suggests that innate immunity in these mucosal tissues played a significant role in the antiviral process. Similarly, in a very recent study conducted by *Kushala et al. (2022)*, significant up-regulation of IL-10, IL-1$\beta$, and TNF-$\alpha$ were detected in the gill of naturally KSD-affected koi. Interestingly, it was found that mucosal immunity plays even more important role in protection when a cohabitation method is used for inducing the infection. For instance, when different Nile tilapia strains were infected with tilapia lake virus (TiLV), the virus load was significantly lower with less mortality in the strains infected through the cohabitation method compared to strains infected with an intraperitoneal injection (*Adamek et al., 2022b*). In our case, despite gill containing the highest virus load, the strong innate immune mucosal response could be seen in gill and gut tissues in carp strains infected through the cohabitation method.

The common carp's ability to defend itself against viral infections is underlined by the overproduction of mucus on skin and gill during CEV infection (*Zhang et al., 2017*). And it has been recorded that skin epidermal erosion following partial ulceration is one of the main sign during CEV infections (*Miyazaki, Isshiki & Katsuyuki, 2005*). In our case, skin despite showing the second highest viral loads of CEV, did not induce strong innate immune response, except IL-1$\beta$, the only gene that was up-regulated in koi carp during post-CEV infection. It seems that other immune genes related to skin protection could play a significant role during the infection with this pathogen. For instance, mucins (a membrane-associated glycoprotein) that are the main components of the mucosal barrier studied during the infections with carp viruses CyHV-3, and SVCV including CEV (genogroup I) (*Adamek et al., 2017a*). The down-regulation of mucin mRNA expression was detected in gill and gut mucosal tissues of carp infected with above cited pathogens.

There is a far paucity of research about T cell markers (CD4, CD8b1 and *GzmA*) to CEV infection in the mucosal tissue of common carp. Gill is the only mucosal organ where down-regulation of gene encoding for CD4 and CD8 cytotoxic T cells markers were detected in koi on day 6 and 9 post-exposure to CEV genogroup IIa infection compared with control as well to the expression in CEV-infected AS carp (*Adamek et al., 2021*). Our findings agree with their study reporting down-regulation of T cell markers such as CD4 and *GzmA* in koi gill on 5 dpi to CEV genogroup IIa compared to control and AS strains. Further, no CD4, CD8b1 and *GzmA* responses were noticed in other mucosal organs, such as gut and skin. Interestingly up-regulation of Mx2 was noticed in all mucosal organs during CEV exposure indicating that mucosal tissues induced a robust antiviral response to CEV infection. The evidence from the previous studies (*Adamek et al., 2014*; *Adamek et al., 2017c*) shows that the antiviral response seemed positively correlated with the virus load in infected fish and cannot be related to the resistance of carp strains to infection. Our study found a similar correlation between virus load and antiviral response in gill tissue. However, the gut harboured the least amount of virus load among the mucosal organs, showed similar up-regulation expression levels of Mx2 in both strains during post-CEV exposure.

Furthermore, mucosal response of mucin-encoding genes (Muc2 and Muc5b) in AS and koi upon CEV genogroup IIa infection in gill, gut and skin tissues was evaluated. In our case tissue-specific expression of mucin genes was observed, where Muc2 was expressed exclusively in the gut tissue, and Muc5b was expressed in the gill and skin tissues. Tissue specific expression of Muc2 and Muc5b genes have been noticed in the previous studies. For instance, Muc2 in carp is predominantly expressed in gut and Muc5b gene, mostly expressed in skin and brancial epithelium (*Van der Marel et al., 2012*). In addition, the gills predominantly express Muc2-like as the secreted mucin when compared to Muc5b, while the skin mainly secretes Muc5b in comparison to Muc2-like (*Lang et al., 2004*). Muc2-like, Muc5b, and the previously described Muc2 from the gut are the primary secreted mucins in the major mucosal tissues of the common carp. The current study revealed a marked down-regulation of Muc5b transcripts in the skin tissue of koi at day 5 post-infection as compared to the control group. These findings suggest that Muc5b may play a critical role in the defense mechanisms of koi against the infectious agent. The reduced expression of

Muc5b transcripts in the skin tissue may lead to impaired mucus production, which may affect the skin barrier function and increase the susceptibility of koi to infection.

## CONCLUSIONS

In conclusion, this is the first study where important immune gene expression was determined in different carp strains infected with CEV genogroup I and compared with responses against genogroup IIa. Fish infected with both genogroups demonstrated similar expression patterns of selected immune-related genes. Interestingly, koi carp was the only strain where most genes showed significant differences in fish infected with both CEV genogroups, with slight variation in the expression pattern. Furthermore, the expression pattern for most genes in KSD-resistant AS strain (*Adamek et al., 2017c*; *Adamek et al., 2021*) resemble some similarities to koi. These similarities might be due to the origin of both carp strains from the same species, such as *Cyprinus rubrofuscus*. According to the observed expression patterns, the difference in susceptibility does not seem to be related to the kinetics of expression of selected immune genes studied in this work. The expression patterns however could help explaining the recorded pathology. Furthermore, up-regulation of mRNA expression of most of the selected immune genes in koi gill and gut tissues suggests potential systemic mucosal response against CEV infection. Ultimately, the implementation of further studies of immune responses against CEV should be under strong consideration because of the paucity of literature regarding the immune responses of carp to CEV infections.

### Funding

This research was supported by Deutsche Forschungsgemeinschaft (DFG project number 426513195) and by the Ministry of Education, Youth and Sports of the Czech Republic—CENAKVA project (LM2018099) and PROFISH project (CZ.02.1.01/0.0/0.0/16_019/0000869). The funders had no role in study design, data collection and analysis, decision to publish, or preparation of the manuscript.

### Grant Disclosures

The following grant information was disclosed by the authors:
Deutsche Forschungsgemeinschaft: DFG project number 426513195).
Ministry of Education, Youth and Sports of the Czech Republic—CENAKVA project: LM2018099.
PROFISH project: CZ.02.1.01/0.0/0.0/16_019/0000869.

### Competing Interests

The authors declare there are no competing interests.

### Author Contributions

- Ali Asghar Baloch performed the experiments, analyzed the data, prepared figures and/or tables, authored or reviewed drafts of the article, and approved the final draft.

- Dieter Steinhagen conceived and designed the experiments, analyzed the data, authored or reviewed drafts of the article, and approved the final draft.
- David Gela conceived and designed the experiments, performed the experiments, authored or reviewed drafts of the article, and approved the final draft.
- Martin Kocour conceived and designed the experiments, performed the experiments, authored or reviewed drafts of the article, and approved the final draft.
- Veronika Piačková conceived and designed the experiments, performed the experiments, analyzed the data, authored or reviewed drafts of the article, and approved the final draft.
- Mikolaj Adamek conceived and designed the experiments, performed the experiments, analyzed the data, prepared figures and/or tables, authored or reviewed drafts of the article, and approved the final draft.

## Animal Ethics

The following information was supplied relating to ethical approvals (*i.e.,* approving body and any reference numbers):

Lower Saxony State Office for Consumer Protection and Food Safety (LAVES), Oldenburg, Germany prodided full approval for this research under the reference number: 33.19–425 2-04-16/2144.

## Data Availability

The raw data is available in the Supplemental Files.

## Supplemental Information

Supplemental information for this article can be found online at http://dx.doi.org/10.7717/peerj.15614#supplemental-information.

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
