# Peer review of "Immune responses in carp strains with different susceptibility to carp edema virus disease"

_PeerJ, doi:10.7717/peerj.15614_

## Round 0.1 · original submission · Major Revisions

To avoid data duplication, the authors need to pay attention to the presentation of data extracted from the earlier studies in Tables 1 and 2. The suggested analytical measurements by reviewer 1 is highly needed.

Reviewer 1 ·

Basic reporting

In the manuscript 80129, Ali Asghar Baloch and co-authors performed experiments to investigate immune responses in four carp strains with different susceptibility to carp edema virus disease. The experiments are associated with four carp strains, multiple post-infection time points, two CEV genogroups, three tissues and the expression level of nine immune-related genes. Overall, the manuscript was well written and organized. The results were well presented. However, some issues especially in validity of the findings required to be improved and supplemented. I suggest to give a major revision.

Experimental design

Overall, the experimental design was good. However, the study was largely based on previously published results especially tables 2&3, I feel concerned about duplicate publication in terms of this part. Authors should focus on the main results in the present study. Relevant tables can be put in supplementary materials. Please pay attention to this issue and handle this problem properly.

Validity of the findings

1.In general, ELISA detection is necessary to validate the expression of cytokines whereas authors just used qPCR to validate the expression level of nine immune-related genes. Authors should provide relevant ELISA detection results of cytokines.
2.It’s not robust to just use qPCR assay for IgM detection.
3.Please confirm the accuracy of the Mean and SD in tables in the manuscript. It seemed that SD is very large in tables.

Additional comments

1.Results description should be presented straightforwardly, description associated with methodology in Results part should be deleted.
2.Some typos existed in this manuscript. For example, line 244 1×105 copies, line 247, reference gene/105.Please check throughout the manuscript.

Reviewer 2 ·

Basic reporting

This study monitored the immune profile of 4 strains of carps challenged by CEV genogroup I and IIa. The experimental procedures got ethical approved by Lower Saxony State Office for Consumer Protection and Food Safety (LAVES). Valuable immune genes expression level of different carp strains infected with CEV genogroup I and IIa were reported in this article. This would contribute to the immune diagnosis and evaluation of potential treatment of CEV.

Experimental design

To check for contaminated DNA (such as genomic DNA), I would recommend a negative control (-RT control) during the RT-PCR process. If PCR amplification is observed in this control, which should not be, it is most likely derived from contaminating DNA. Also, if possible, positive control for these 9 genes (cDNA) is preferred.

Validity of the findings

The raw data is provided. They are robust, statistically sound

Reviewer 3 ·

Basic reporting

This study examined immune gene expression in mucosal tissues for clues related to differential susceptibility to viral infection between different carp strains. This is a valuable attempt to study breeding-related traits from mucosal immunity. However, the introduction to the background of fish mucosal immunity is not sufficient. Writing is bad for logic.
(1) Abstract: Since the conclusions are about the mucosal immune response, why not examine the key effector mucins at the mucosa surface? It has been documented that viral infection, including CEV, reduces the expression of gene coding membrane-associated glycoproteins mucin, thus, qPCR analysis of MUC genes and Alician blue staining could be included.
(2) The involvement of T cell responses should be mentioned in the introduction.
(3) For the presentation of all qPCR results (Figures 1-6), I recommend using relative fold change corrected through the expression of housekeeping gene, rather than the copy number.
(4) Line 260: Move the introduction of those previously published CEV infection trials to the introduction section and provide more details.
(5) Line 241 & 244: The numbers "0", "6", “5” after "10" should be superscripted.

Experimental design

The current experimental design is very simple.
(1) Line 96: Why use fish of different body weights? For the animal experiment, the experimental setup should be demonstrated in a diagram.
(2) In Materials & Methods, there is a lack of a separate section describing the sampling procedure.

Validity of the findings

(1) A MX2 alone is not enough to understand the antiviral response, because the virus-induced interferon and interferon-induced genes also play important roles. Please refer to the references titled "Carp edema virus and immune response in carp (Cyprinus carpio): Current knowledge" (published in Journal of Fish Disease) and "Acute Infection of Viral Pathogens and Their Innate Immune Escape" (published in Frontiers in Immunology).
(2) In the Discussion, the suppressed expression of immune genes (CD4 and CD8) may be due to viral immunomodulation or related escape from the host immune system. Therefore, this point should be discussed.
(3) It is better to discuss genes associated with anti-viral response, separating from the mucosal effector. IgM and mucin work together at the surface.
(4) There is a lack of discussion on interplay or crosstalk between multiple mucosal surfaces. See the reference titled "Recent Insights into Cellular Crosstalk in Respiratory and Gastrointestinal Mucosal Immune Systems" published in Immune Network.
(5) Line 475-502: Here, the discussion about the expression of Th and cytotoxic T cell related genes can be explained. There are more result description but no much biological significance explained.
(6) Some points are over-discussed, such as lines 503-504 and Line 532. Discussion should be supported by data or references.

---

## Round 0.2 · accepted · Accept

The authors have properly responded to all reviewer's comments. The manuscript could be accepted in its present form.

Reviewer 2 ·

Basic reporting

no comment

Experimental design

It is addressed well

Validity of the findings

No comment